# Baby Food Jars as a Dietary Source of Essential (K, Na, Ca, Mg, Fe, Zn, Cu, Co, Mo, Mn) and Toxic Elements (Al, Cd, Pb, B, Ba, V, Sr, Li, Ni)

Santiago González-Suárez [1,2], Soraya Paz-Montelongo [1,2,*], Daniel Niebla-Canelo [1,2], Samuel Alejandro-Vega [1,2], Dailos González-Weller [2,3], Carmen Rubio-Armendáriz [1,2], Arturo Hardisson [1,2] and Ángel J. Gutiérrez-Fernández [1,2]

1   Área de Toxicología, Universidad de La Laguna, 38071 La Laguna, Islas Canarias, Spain
2   Grupo Interuniversitario de Toxicología Alimentaria y Ambiental, Universidad de La Laguna, 38071 La Laguna, Islas Canarias, Spain
3   Servicio Canario de Salud Pública, Laboratorio Central, 38001 Santa Cruz de Tenerife, Islas Canarias, Spain
*   Correspondence: spazmont@ull.edu.es; Tel.: +34-634541612

**Abstract:** Baby food from jars is made of meat, vegetables or fruits, and might be a valuable source of essential elements such as Na or K. However, these infant products could also be a source of toxic elements such as Al or Cd, which are dangerous to infants. In total, 45 samples of various kinds of baby food in jars (meat, vegetables, fruit and mixed) were analyzed using inductively coupled plasma spectrometry (ICP OES) with the aim of evaluating the daily intake of essential elements (K, Na, Ca, Mg, Fe, Zn, Cu, Co, Mo, Mn) and dietary exposure to toxic elements (Al, Cd, Pb, B, Ba, V, Sr, Li, Ni). Mixed jars registered the highest concentrations of Na, Ca, Zn, Fe. Al ($8.22 \pm 8.97$ mg/kg wet weight) stands out in vegetable jars. In total, 130 g/day of mixed jars fulfills daily Zn and Ca requirements. These consumption scenarios (130 g/day, 250 g/day) supposed high Mn intakes (40 times higher than the recommended value), which could pose a risk to infants' health. Pb, Ni, Cd and Al intakes exceed the maximum values. It is recommended to avoid the daily consumption of these products since it can pose a risk to the health of infants. Chemical compounds studied in this article: Nitric acid (PubChem: CID 944).

**Keywords:** baby food in jars; risk assessment; toxic risk; toxic elements

## 1. Introduction

Newborns are fed with breast milk. Breast milk is an essential food for newborns because it provides essential elements in suitable quantities for their development and it contains other substances such as digestive enzymes, bifidogenic factors and growth factors [1]. The World Health Organization [2] recommends feeding babies with breast milk for at least the first 6 months of their life. As the baby grows, the requirements of macronutrients and essential elements increase, and it necessary to include other foods into the baby's diet. The incorporation of different foods throughout the growth of the infant follows an established calendar attending to the stages of baby growth [3–5]. Complementary food is made up of cereals, dairy products and well-known baby food in jars [6].

Baby food in jars is made of fruits, meat, fish or vegetables that are presented in puree form with a semi-solid consistency. Baby food in jars is packed in containers with hermetic closures that are subjected to thermal process to ensure their preservation. In addition, these products are free of preservatives or artificial pigments. Baby food in jars is an easy-access product with great approval from the population. Taking into account that these products are made from fruits, vegetables, meat or fish, it is expected that they provide essential elements (K, Na, Ca, Mg, Fe, Zn, Cu, Co, Mo, Mn) required daily for the development of babies [7,8].

K participate in synthesis processes of protein and glycogen, in transmission of the nervous and muscular impulse [9], both K and Na are part of the $Na^+K^+$-ATPase pump [9–11]. Ca is necessary for the bone structure and function [12,13]. Mg is a cofactor that participates in more than 300 enzymatic reactions [9] and participates in important metabolic pathways such as blood coagulation [14,15].

Fe is involved in oxygen transport because is part of hemoglobin, also participates in mitochondrial respiration, DNA synthesis or in the inactivation of free radicals [16–18]. Zn participates in regulation of gene expression [9,19–22] and in the immune system function [23]. Cu is necessary in the metabolism of iron, in the regulation of gene expression and in the mitochondrial function [9,14]. Co is a component of cobalamin or vitamin B12, which is necessary for the proper function of the brain, nervous system, protein synthesis and DNA regulation [9,24,25]. Mo is an important cofactor of metalloenzymes involved in the catabolism of purines, pyridines and sulphur-containing amino acids [9,26,27]. Mn is part of enzymes such as peptidases, phosphatases, arginase, phosphoglucomutase and glucosyl transferases [9], and is necessary for the amino acids, cholesterol and carbohydrates metabolism [14].

However, although the baby food quality requirements are strict, it is necessary to consider that, due to the ingredients that these prepared foods contain, it could be a source of toxic elements (Al, Cd, Pb, B, Ba, V, Sr, Li, Ni) dangerous for the babies' health [7,8,28,29].

Al is a neurotoxic element, which tends to accumulate in the bones, kidneys, liver and brain. Long-term exposure to high Al levels is associated with Alzheimer's, and it could also interfere with essential elements such as Ca [30–33]. Cd also affects Ca homeostasis and therefore leads to cardiovascular problems. This metal could also damage the renal tubules, which are implicated in nutrient reabsorption mechanisms [34–36].

Pb is a neurotoxic element as it causes serious damage to the central nervous system (CNS), especially in children and fetuses [37–40]. Moreover, Pb can lead to kidney diseases and disorders of the gastrointestinal tract [32].

Excessive intake of B in experimental animals shows adverse effects on growth and reproductive function [14]. High Ba intake can cause tachycardia, hypertension, hypotension, muscle weakness and paralysis, because this element increases the intracellular potassium levels [41]. High V intake causes gastrointestinal disorders [14]. Excessive Sr intakes can cause phosphorus deficiency, and its accumulation in bones could lead to increases in the bone density [42]. Although cases of Li poisoning through food are unknown, high intakes can cause altered consciousness, ataxia, nausea, tremors, apathy, polyuria, vomiting, myopathy [43,44]. Ni especially affects individuals with Ni-sensitivity or kidney problems [14,45].

It is for this reason that food safety agencies such as the European Food Safety Authority have set different reference values for maximum and recommended intakes (Table 1).

Because of the high consumption of baby food in jars by the infant population, and since babies are a vulnerable group to the toxic effects of certain elements, it is necessary to determine the content of essential elements for their development, as well as toxic elements that pose a health risk.

The objectives of the present study are (i) to determine the content of essential elements (K, Na, Ca, Mg, Fe, Zn, Cu, Co, Mo, Mn) and toxic elements (Al, Cd, Pb, Cr, B, Ba, V, Sr, Li, Ni) in baby food in jars, (ii) to study the existence of significant differences between the different types of baby food in jars analyzed, (iii) to evaluate the dietary intake of essential elements considering the daily requirements of the infant population and (iv) to assess the dietary exposure to toxic elements (v) to determine if the consumption of these products constitutes a health risk for infants.

**Table 1.** Guideline values.

| Elements | | Guideline Values | Reference |
|---|---|---|---|
| Ca | | 280 mg/day | [46] |
| Cu | | 0.4 mg/day | [47] |
| K | | 750 mg/day | [48] |
| Mg | AI | 80/mg/day | [49] |
| Mn | | 0.02–0.05 mg/day | [50] |
| Mo | | 10 µg/day | [27] |
| Na | | 0.2 g/day | [51] |
| Zn | PRI | 2.9 mg/day | [20] |
| Fe | | 11 mg/day | [52] |
| Al | TWI | 1 mg/kg body weight/week | [53] |
| B | UL | 0.16 mg/kg body weight/day | [54] |
| Ba | TDI | 0.2 mg/kg body weight/week | [41] |
| Cd | TWI | 2.5 µg/kg body weight/week | [55] |
| Ni | TDI | 2.8 µg/kg body weight/day | [45] |
| Pb | BMDL | 0.5 µg/kg body weight/day | [56] |
| Sr | UL | 0.13 µg/kg body weight/day | [42] |
| V | | 1.8 mg/day | [14] |

AI = Adequate intake; PRI = Population reference intake; TWI = Tolerable weekly intake; RDI = Reference daily intake; TDI = Tolerable daily intake; BMDL= Benchmark dose (lower confident unit); UL= Tolerable upper intake level.

## 2. Material and Methods

The treatment and analysis of the samples were performed by using analytical grade chemical reagents and high-purity distilled water obtained from the Milli-Q water purification system (Millipore, MA, USA). Likewise, the material used was washed with Acationox laboratory detergent (Merck, Germany) and distilled water [31].

### 2.1. Samples and Treatment

A total of 45 samples of different baby food in jar types were analyzed (Table 2). The analyzed samples were acquired in supermarkets and large commercial stores from Tenrife, Spain, and were analyzed before their expiration date.

**Table 2.** Descriptive parameters of the analyzed samples.

| | Baby Food Jar Types | Number of Samples | Package |
|---|---|---|---|
| Fruits | Mixed fruits | 5 | |
| | Apple | 5 | |
| | Banana | 5 | |
| Meat | Chicken stew | 10 | Glass jar |
| Vegetables | Mixed vegetables boiled | 10 | |
| Mixed | Gardener-style tender veal | 10 | |

Overall, 10 g of each sample were weighed, in triplicate, in porcelain capsules (Staatlich, Berlin, Germany), previously homogenized. Samples were placed in an oven (Nabertherm, Lilienthal, Germany) at 70 °C for 24 h for drying. Subsequently, the samples were subjected to acid digestion by adding about 5 mL of 65% $HNO_3$ (Sisgma Aldrich, Taufkirchen, Germany) until evaporation of the nitric acid with a heating plate (Nabertherm, Lilienthal, Germany). The samples were then incinerated in a muffle furnace (Nabertherm, Lilienthal, Germany) at 450 °C $\pm$ 25 °C throughout 24 h, applying a temperature ramp of 50 °C per hour. The ashes obtained were dissolved in 1.5% $HNO_3$ solution (Sigma Aldrich,

Taufkirchen, Germany) up to a total volume of 25 mL in a volumetric flask [57]. The solutions were transferred to polyethylene containers for further analysis (Figure 1).

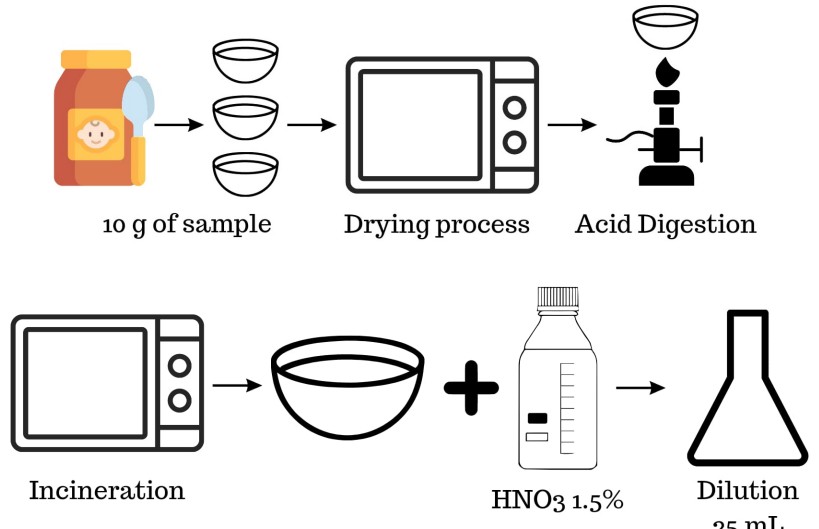

**Figure 1.** Experimental procedure of mineralization by incineration.

## 2.2. Analytical Method and Quality Control

Essential elements (K, Na, Ca, Mg, Fe, Zn, Cu, Co, Mo, Mn) and toxic elements (Al, Cd, Pb, B, Ba, V, Sr, Li, Ni) were determined by inductively coupled plasma optical emission spectrometer (ICP-OES) model ICAP 6300 Duo Thermo Scientific (Waltham, MA, USA) with an auto sampler (CETAX model ASX-520). It is a technique frequently used in the determination of elements in different food and biological matrices [7,58–60], and it is one of the most sensitive and precise techniques of instrumental analysis, just behind ICP-MS [60].

The instrumental ICP-OES conditions were: sample injection to the 50 rpm flow pump; stabilization time of 0 s, approximate RF power of 1150 W and gas flow (nebulizer gas flow, auxiliary gas flow) of 0.5 L/min [31,61]. Instrumental wavelengths (nm) of the analyzed elements. Instrumental quantification limits (LOQ) of the analyzed elements were calculated as 10 times the standard deviation (SD), resulting from the analysis of 15 targets under reproducibility conditions [62] (Table 3).

Calibrations were performed using certified standard solutions. Specifically, for the metals Na, Ca, K and Mg, the certified standard IV-STOCK-2 of Inorganic Ventures was used, with a certified concentration for each of the metals of 10,000 µg/mL and for the rest of the metals (V, Mn, Fe, Cu, Zn, Cr, Mo, Co, B, Ba, Li, Sr, Ni, Si, Al, Pb and Cd), the certified standard Multi-Element Std, SCP28AES from SCP Science, with a certified concentration for each of the metals of 100 mg/L, has been used.

From these, and for each of the metals analyzed in this study, the different concentrations of the calibration standards for the preparation of the calibration lines were prepared, all of them in sufficient quantity for 100 mL in 1.5% nitric acid.

Table 4 shows the recovery study performed with Certified Reference Materials (CRM) in order to check the accuracy and precision of the analytical method. The CRM used, from the National Institute of Standards and Technology (NIST), were: SRM 1515 Apple Leaves, SRM 1548a Typical Diet, SRM 1577b Bovine Liver and SRM 1567a Wheat Flour. The standard additions method was used in the case of Li, by adding known quantities of Li to dehydrated samples of baby food from jars. The recovery percentages obtained were good (over 92%).

**Table 3.** Instrumental parameters and quantification limits of the analysis method.

| Elements | Instrumental Wavelengths (nm) | LOQ (mg/L) |
|---|---|---|
| Al | 167.0 | 0.012 |
| B | 249.7 | 0.012 |
| Ba | 455.4 | 0.005 |
| Ca | 317.9 | 1.995 |
| Cd | 226.5 | 0.001 |
| Co | 228.6 | 0.002 |
| Cu | 327.3 | 0.012 |
| Fe | 259.9 | 0.009 |
| K | 769.9 | 1.884 |
| Li | 670.8 | 0.013 |
| Mg | 279.1 | 1.943 |
| Mn | 257.6 | 0.008 |
| Mo | 202.0 | 0.002 |
| Na | 589.6 | 3.655 |
| Ni | 231.6 | 0.003 |
| Pb | 220.3 | 0.001 |
| Sr | 407.7 | 0.003 |
| V | 310.2 | 0.005 |
| Zn | 206.2 | 0.007 |

**Table 4.** Recovery percentage found for the reference materials used.

| Metal | Material | Concentration Found (mg/kg) | Certified Concentration (mg/kg) | R (%) |
|---|---|---|---|---|
| Na | | $0.242 \pm 0.006$ | $0.238 \pm 0.010$ | 98.3 |
| K | SRM 1577b Bovine Liver | $0.994 \pm 0.002$ | $0.936 \pm 0.007$ | 94.2 |
| Ca | | $116 \pm 4$ | $111.1 \pm 8.5$ | 95.8 |
| Mg | | $601 \pm 28$ | $559.5 \pm 46$ | 93.1 |
| Al | | $286 \pm 9$ | $285.1 \pm 26$ | 99.7 |
| B | SRM 1515 Apple Leaves | $27.0 \pm 2.0$ | $27.0 \pm 1.5$ | 99.9 |
| Mo | | $0.09 \pm 0.01$ | $0.09 \pm 0.02$ | 99.4 |
| Sr | | $25.0 \pm 2.0$ | $24.6 \pm 4.0$ | 98.3 |
| Ba | | $1.10 \pm 0.10$ | $1.13 \pm 0.09$ | 102.5 |
| Ni | SRM 1548a Typical Diet | $0.37 \pm 0.02$ | $0.38 \pm 0.04$ | 102.3 |
| Pb | | $0.044 \pm 0.000$ | $0.044 \pm 0.013$ | 98.9 |
| Cd | | $0.026 \pm 0.002$ | $0.026 \pm 0.008$ | 98.4 |
| Co | | $0.006 \pm 0.00$ | $0.006 \pm 0.002$ | 102.4 |
| Cu | | $2.1 \pm 0.2$ | $2.09 \pm 0.4$ | 99.7 |
| Fe | SRM 1567a Wheat Flour | $14.1 \pm 0.5$ | $13.9 \pm 0.3$ | 98.9 |
| Mn | | $9.4 \pm 0.9$ | $9.6 \pm 1.5$ | 102.4 |
| V | | $0.011 \pm 0.00$ | $0.011 \pm 0.00$ | 99.4 |
| Zn | | $11.6 \pm 0.4$ | $11.9 \pm 0.2$ | 102.7 |
| Li | Standard Addition Method | $0.2 \pm 0.02$ | $0.19 \pm 0.03$ | 95.0 |

The criteria, from an analytical point of view, that guarantee the correct evaluation of the results presented in the work have been validated, with the parameters of Accuracy (established as recovery), Precision (established as reproducibility) and Specificity (it has been carried out by checking that the method is free of spectral interferences for each of the metals studied) applied to the reference materials used in the study.

In addition to these, the following parameters were also monitored and evaluated:

- The sensitivity of the calibration line: this was carried out with the response of the lowest standard of the calibration lines for each of the metals, setting as an acceptance criterion a response 3 times higher than the response of the blank (1.5% $HNO_3$ solution).
- The linearity of the calibration line: this was established by applying the relative calibration errors, setting as acceptance criteria a maximum of 15% of this error for all metals in the lowest standards of each line and 10% in the rest of the points of the calibration lines.
- The repeatability precision of the method: it was established with samples analyzed in duplicate, setting as acceptance criterion the following formula: $r = 2\sqrt{2} \cdot Sr$, where Sr is the repeatability standard deviation.

Finally, when measuring each sample, two replicates were analyzed, obtaining from each of them a mean concentration and a % RSD value for the quantifiable intervals of the method. To take a measurement as valid, a %RSD $\leq$ 10% was established.

### 2.3. Statistical Analysis

Statistical analysis was performed using the IBM SPSS STATISTICS 24.0 for Mac™ program (IBM, Magdeburg, Germany). The statistical study was carried out to study the existence of significant differences between the different types of baby food in jars (vegetables, fruits, meat or mixed).

A Kolmogorov–Smirnov test was used to check the data normality, and the Levene statistic for the variances homogeneity [63]. As the data obtained did not follow a normal distribution, both a Kruskal–Wallis test and the Mann–Whitney U test were applied to validate the existence of significant differences ($p < 0.05$), as they are both non-parametric tests [64].

### 2.4. Dietary Intake Calculations

Dietary intake assessment is based on the estimated daily intake (EDI), which is the quantity of essential elements or toxic elements ingested with a serving of baby food jar. It is calculated as follows:

$$\text{EDI}\left(\frac{\text{mg}}{\text{day}}\right) = \text{Element concentration}\left(\frac{\text{mg}}{\text{kg}}\right) \times \text{Baby food jar}\left(\frac{\text{kg}}{\text{day}}\right)$$

Once the EDI values were obtained for each analyzed element, the percentages of contribution (%) to the values of recommended daily intake or maximum permissible intakes were obtained. They are calculated as shown below:

$$\text{Contribution (\%)} = \left(\frac{\text{EDI}}{\text{Guideline values}}\right) \times 100$$

The margin of exposure (MOE) is employed by risk assessors to examine potential safety concerns derived from substances which are both genotoxic (they may damage DNA) and carcinogenic present in food and feed. It is a ratio of two factors which assesses the dose at which measurable adverse effect is first observed for a given population, and the level of exposure to the substance considered.

Additionally, is interpreted as follows: "The Scientific Committee is of the view that in general a margin of exposure of 10,000 or higher, if it is based on the $\text{BMDL}_{10}$ from an animal carcinogenicity study, and taking into account overall uncertainties in the interpretation, would be of low concern from a public health point of view" [65]. Additionally, it is determined as:

$$\text{MOE} = \frac{\text{BMDL}\left(\frac{\text{mg}}{\text{kg b.w.} \times \text{day}}\right) \times \text{B.W. (kg)}}{\text{IDE}\left(\frac{\text{mg}}{\text{day}}\right)}$$

## 3. Results and Discussion

### 3.1. Essential and Toxic Elements Concentrations in Baby Food in Jars

Table 5 shows the mean concentrations (mg/kg wet weight) and standard deviations (SD) of the samples analyzed according to the type of jar.

In the case of essential elements, the mean content of Na (2464 $\pm$ 106 mg/kg ww) and Ca (921 $\pm$ 62.7 mg/kg ww) recorded in mixed jars stands out. On the other hand, the highest average levels of K (4213 $\pm$ 548 mg/kg ww) and Mg (319 $\pm$ 47.9 mg/kg ww), were found in meat and vegetables jars, respectively. It should be noted the mean content of Zn (18.6 $\pm$ 1.84 mg/kg ww) and Fe (10.5 $\pm$ 0.76 mg/kg ww) found in mixed jars.

**Table 5.** Concentration (mg/kg wet weight) and standard deviation (SD) of the analyzed metals in the samples of baby food jar.

| Element | Concentration (mg/kg Wet Weight) $\pm$ SD | | | |
| --- | --- | --- | --- | --- |
| | Fruits | Vegetables | Meat | Mixed |
| Ca | $221 \pm 78.2$ | $456 \pm 38.1$ | $798 \pm 151$ | $921 \pm 62.7$ |
| K | $3882 \pm 1178$ | $3686 \pm 363$ | $4213 \pm 548$ | $4066 \pm 1000$ |
| Na | $425 \pm 170$ | $752 \pm 184$ | $1943 \pm 740$ | $2464 \pm 106$ |
| Mg | $307 \pm 97.3$ | $319 \pm 47.9$ | $302 \pm 96.0$ | $316 \pm 47.5$ |
| Al | $8.14 \pm 5.56$ | $8.22 \pm 8.97$ | $5.89 \pm 6.83$ | $5.46 \pm 4.32$ |
| Cd | $0.01 \pm 0.002$ | $0.03 \pm 0.004$ | $0.03 \pm 0.006$ | $0.02 \pm 0.004$ |
| Pb | $0.11 \pm 0.04$ | $0.14 \pm 0.04$ | $0.17 \pm 0.05$ | $0.16 \pm 0.03$ |
| Ba | $1.09 \pm 0.42$ | $1.75 \pm 0.39$ | $1.14 \pm 0.29$ | $1.61 \pm 0.62$ |
| B | $5.37 \pm 2.13$ | $2.40 \pm 1.06$ | $1.73 \pm 0.80$ | $1.91 \pm 0.78$ |
| Co | $0.02 \pm 0.0004$ | $0.02 \pm 0.0005$ | $0.02 \pm 0.02$ | $0.02 \pm 0.003$ |
| Cu | $2.70 \pm 0.77$ | $1.71 \pm 0.49$ | $1.74 \pm 0.58$ | $1.40 \pm 0.34$ |
| Sr | $3.55 \pm 3.39$ | $4.34 \pm 0.43$ | $2.64 \pm 0.42$ | $4.10 \pm 1.46$ |
| Fe | $5.42 \pm 1.24$ | $8.40 \pm 0.83$ | $7.83 \pm 1.79$ | $10.5 \pm 0.76$ |
| Li | $307 \pm 97.3$ | $319 \pm 48.0$ | $302 \pm 96.0$ | $316 \pm 47.5$ |
| Mn | $3.20 \pm 1.97$ | $2.93 \pm 0.36$ | $2.28 \pm 0.69$ | $2.28 \pm 0.27$ |
| Mo | $0.06 \pm 0.01$ | $0.18 \pm 0.05$ | $0.15 \pm 0.02$ | $0.12 \pm 0.01$ |
| Ni | $0.13 \pm 0.04$ | $0.87 \pm 0.07$ | $0.17 \pm 0.03$ | $0.17 \pm 0.02$ |
| V | $0.16 \pm 0.07$ | $0.19 \pm 0.08$ | $0.14 \pm 0.08$ | $0.13 \pm 0.08$ |
| Zn | $2.90 \pm 0.49$ | $6.15 \pm 1.02$ | $12.5 \pm 1.54$ | $18.6 \pm 1.84$ |

Regarding the toxic elements, high Li levels were registered in all the analyzed types, highlighting the average level of Li ($319 \pm 48.0$ mg/kg ww) found in meat jars. Likewise, the Al content ($8.22 \pm 8.97$ mg/kg ww) recorded in vegetable jars were the highest, which may be due to the fact that vegetables absorb and accumulate higher levels of Al from the soil [32,66]. While the highest Cd ($0.03 \pm 0.006$ mg/kg ww) and Pb ($0.17 \pm 0.05$ mg/kg ww) levels were found in meat jars.

COMMISSION REGULATION (EU) 2021/1317 of 9 August 2021 amending Regulation (EC) No 1881/2006 as regards the maximum Pb content in certain food products, sets a maximum limit for Pb in food children of 0.020 mg/kg of fresh weight [67], considering this maximum limit all samples groups analyzed exceeds it between a 550% and 850%.

The European Regulation (EC) No. 1881/2006, of 19 December 2006, as amended by COMMISSION REGULATION (EU) No. 2021/1323 of 10 August 2021, establishes the maximum admissible concentrations in processed foods based on cereals and infant food for infants and young children, with the maximum Cd level of 0.04 mg/kg of fresh weight [68]. Considering the data obtained in the present study, the samples analyzed are below this limit.

Significant differences ($p < 0.05$) were detected by using statistical analysis in Ca and Na content between meat and mixed jars with those of vegetables and fruits revealed. As well as were found statistical differences in Fe levels between the meat and vegetable jars with those of fruits, and these with the mixed ones. Likewise, the statistical study has shown statistical differences ($p < 0.05$) in the content of Cd and Pb between the meat, mixed and vegetable jars with the fruit jars.

The Ba content is statistically different ($p < 0.05$) between the meat and fruit jars with those of vegetables and mixed. In the case of Cu, Mo, Ni and B, it is statistically differentiated between fruit jars with the rest of the types. On the other hand, the registered levels of Zn turned out to be significantly different between the four types of baby food in jars analyzed. Finally, the Sr content is statistically different between meat and vegetable jars.

The differences found between the different types are mainly due to their composition, since the levels of essential and toxic elements in vegetables and fruits are influenced by factors such as climate, soil pH, soil composition, irrigated water, sources from nearby pollution, used pesticides and fertilizers [69,70], whereas in meats, the content of these elements will be influenced by the age of the animal, species and diet [71–73].

### 3.2. Comparison with Other Authors

Table 6 shows the values obtained by other authors in samples of baby food in jars.

In fruit-based jars, the concentrations found by Chekri et al. (2019) of Ba, Ni, V, Al, Cd and Sr are lower than those recorded in the present study for the fruit jars analyzed [8]. On the other hand, the average Co level (0.00287 ± 0.00104 mg/kg ww) found by this author in fruit jars were slightly higher than the Co value (0.02 ± 0.0004 mg/kg ww) recorded in the present study. The same occurs with the Cd mean level (0.00049 ± 0.00062 mg/kg ww) found by Jean et al. (2018) and Pb mean level (0.00215 ± 0.00208 mg/kg ww) determined by Guérin et al. (2017); in both cases, the values were lower than those found in the present study [28,29].

In the case of meat and vegetable jars, the values found by Chekri et al. (2019) of Al, Ba, Cd, Co, Ni, Sr and V were lower than those recorded in the present study [8]. Moreover, the mean Cd and Pb values found by the cited authors were lower than those recorded in this study [28,29].

The different origin of raw materials, as mentioned above. As well as the increase in pollution, among other causes, may be the reason for this increase in concentrations. Since over the years, the levels of some toxic elements have increased. As it has been reflected in numerous articles and by concern of the European institutions. These studies also show how pollution can migrate from the environment to food. Thus, affecting humans through the food chain [20,46–53,55,74–83]. However, the concentration of some of them is expected to decrease over time [78].

### 3.3. Dietary Exposure Assessment

The dietary exposure assessment was performed considering different consumption scenarios (130 g/day and 250 g/day) and the average weight of 9 kg for infants aged 7–11 months [84]. Likewise, an average consumption of 130 g/day and 250 g/day has been considered, since they are the most common presentations of baby food in jars, being packaged with these quantities.

As for the essential elements (Table 7), an intake of 130 g/day of mixed jars contributes considerably to the recommended daily intake values of elements such as Zn (83.4%) and Ca (42.9%). On the other hand, consuming 130 g/day of the vegetable jars makes a significant contribution to the recommended intakes of Cu (87.5%) and Mg (51.9%). Lastly, 130 g/day of fruit jars signifies a notable contribution of Mo (80.0%).

However, it should be noted that, in almost all cases, the intake of both 130 g/day and 250 g/day of any of the baby food in jars analyzed represents very high percentages over the recommended values. Thus, for example, the case of Mn stands out, with intakes up to 40 times higher than the recommended value or Mo, and with contribution percentages 4 times higher than the maximum. This fact may pose a risk to the health of infants, since it is known that Mn is an element that, in high concentrations, is neurotoxic, affecting the central nervous system and causing an increase in the concentration of blood, due to its narrow relationship with Fe, as well as muscle weakness or lack of motor coordination [14,85]. On the other hand, high intakes of Mo cause effects on the reproductive system, although these effects have been observed in experimental animals [14].

In the case of toxic elements (Table 8), the consumption of 130 g/day and 250 g/day of any of the analyzed baby food from jars poses a high risk for excessive intake of Pb with MOE of 112.5–450 [65,86], of Ni with percentages of contribution of 79.4–873% of the TDI (2.8 μg Ni/kg bw/day) [45], of Cd with percentages of 31.1–250% of TWI (tolerable weekly intake) (2.5 μg Cd/kg bw/day) [55], of Al with percentages of contribution of 55.2–160% of TWI (1 mg Al/kg bw/week) [53] and of B with percentages of contribution of 30.03–93.23% of UL (0.16 mg/kg body weight·day) [54]. The toxic effects of these elements, especially to the infant population, could be serious. The consumption of the analyzed baby food from jars (130 g/day and 250 g/day) cannot be considered safe for babies (9 kg of body weight) aged between 6 or 7 and 11 months.

**Table 6.** Comparison of the obtained results with other authors.

| Reference | Type | Al | Ba | Cd | Co | Ni | Pb | Sr | V |
|---|---|---|---|---|---|---|---|---|---|
| | | | | | | Element Concentration (mg/kg ww) | | | |
| Present study, 2022 | Fruits | 8.14 ± 5.56 | 1.09 ± 0.42 | 0.01 ± 0.002 | 0.02 ± 0.0004 | 0.13 ± 0.04 | 0.11 ± 0.04 | 3.55 ± 3.39 | 0.16 ± 0.07 |
| [8] | | 0.556 ± 0.254 | 0.184 ± 0.057 | 0.00066 ± 0.00049 | 0.00287 ± 0.00104 | 0.0547 ± 0.0267 | - | 0.273 ± 0.150 | 0.00140 ± 0.00081 |
| [29] | | | - | 0.00049 ± 0.00062 | | | | - | |
| [28] | | | | - | | | 0.00215 ± 0.00208 | | - |
| Present study, 2022 | Meat | 5.89 ± 6.83 | 1.14 ± 0.29 | 0.03 ± 0.006 | 0.02 ± 0.02 | 0.17 ± 0.03 | 0.17 ± 0.05 | 2.64 ± 0.42 | 0.14 ± 0.08 |
| [8] | Meat/Fish | 0.597 ± 0.436 | 0.286 ± 0.141 | 0.00931 ± 0.00433 | 0.00382 ± 0.00132 | 0.0757 ± 0.0257 | - | 0.580 ± 0.203 | 0.00256 ± 0.00132 |
| [29] | | | - | 0.00926 ± 0.00448 | | | | - | |
| [28] | | | | - | | | 0.00313 ± 0.00289 | | - |
| Present study, 2022 | Vegetables | 8.22 ± 8.97 | 1.75 ± 0.39 | 0.03 ± 0.004 | 0.02 ± 0.0005 | 0.87 ± 0.07 | 0.14 ± 0.04 | 4.34 ± 0.43 | 0.19 ± 0.08 |
| [8] | | 0.575 ± 0.511 | 0.337 ± 0.316 | 0.00926 ± 0.00448 | 0.00369 ± 0.00267 | 0.0715 ± 0.028 | - | 0.568 ± 0.157 | 0.00219 ± 0.00127 |
| [29] | | | - | 0.00931 ± 0.00433 | | | | - | |
| [28] | | | | - | | | 0.00343 ± 0.00201 | | - |

**Table 7.** Estimated daily intake (mg/day) and percentages of contribution to the AI of the studied essential elements considering a consumption of 130 g/day and 250 g/day of baby food from jars.

| | Guideline Values [20,27,46–52] | Fruits EDI (mg/Day) 130 g/Day | 250 g/Day | Fruits % AI 130 g/Day | 250 g/Day | Vegetables EDI (mg/Day) 130 g/Day | 250 g/Day | Vegetables % AI 130 g/Day | 250 g/Day | Meat EDI (mg/Day) 130 g/Day | 250 g/Day | Meat % AI 130 g/Day | 250 g/Day | Mixed EDI (mg/Day) 130 g/Day | 250 g/Day | Mixed % AI 130 g/Day | 250 g/Day |
|---|---|---|---|---|---|---|---|---|---|---|---|---|---|---|---|---|---|
| Ca | 280 mg/day | 28.7 | 55.3 | 10.3 | 19.8 | 59.3 | 114 | 21.2 | 40.7 | 104 | 200 | 37.1 | 71.4 | 120 | 230 | 42.9 | 82.1 |
| Cu | 0.4 mg/day | 0.35 | 0.68 | 87.5 | 170 | 0.22 | 0.43 | 55.0 | 108 | 0.23 | 0.44 | 57.5 | 110 | 0.18 | 0.35 | 45.0 | 87.5 |
| Fe | 11 mg/day | 0.71 | 1.36 | 6.45 | 12.4 | 1.09 | 2.10 | 9.91 | 19.1 | 1.02 | 1.96 | 9.27 | 17.8 | 1.37 | 2.63 | 12.5 | 23.9 |
| K | 750 mg/day | 505 | 971 | 67.3 | 130 | 479 | 922 | 63.9 | 123 | 548 | 1053 | 73.1 | 140 | 529 | 1017 | 70.5 | 136 |
| Mg | 80 mg/day | 40.0 | 76.8 | 50.0 | 96.0 | 41.5 | 79.8 | 51.9 | 99.8 | 39.3 | 75.5 | 49.1 | 94.4 | 41.1 | 79.0 | 51.4 | 98.8 |
| Mn | 0.02–0.5 mg/day | 0.42 | 0.80 | 2100–84.0 | 4000–160 | 0.38 | 0.73 | 1900–76 | 3650–146 | 0.30 | 0.57 | 1500–60 | 2850–114 | 0.30 | 0.57 | 1500–60 | 2850–114 |
| Mo | 10 µg/day | 0.008 | 0.02 | 80 | 200 | 0.02 | 0.05 | 230 | 500 | 0.02 | 0.04 | 200 | 400 | 0.02 | 0.03 | 200 | 300 |
| Na | Not established | | | | | | | | | - | | | | | | | |
| Zn | 2.9 mg/day | 0.38 | 0.73 | 13.1 | 25.0 | 0.80 | 1.54 | 27.6 | 53.1 | 1.63 | 3.13 | 56.2 | 108 | 2.42 | 4.65 | 83.4 | 160 |

**Table 8.** Estimated daily intake (mg/day) and percentages of contribution to the tolerable weekly intake (TWI), tolerable daily intake (TDI) or upper level intake (UL) of the studied toxic elements considering a consumption of 130 g/day and 250 g/day of baby food from jars.

| | | | Fruits | | | | Vegetables | | | | Meat | | | | Mixed | | | |
|---|---|---|---|---|---|---|---|---|---|---|---|---|---|---|---|---|---|---|
| | Guideline Values | Parameter | EDI (Mg/Day) | | % TWI, TDI or UL | | EDI (Mg/Day) | | % TWI, TDI or UL | | EDI (Mg/Day) | | % TWI, TDI or UL | | EDI (Mg/Day) | | % TWI, TDI or UL | |
| | | | 130 g/Day | 250 g/Day | 130 g/Day | 250 g/Day | 130 g/Day | 250 g/Day | 130 g/Day | 250 g/Day | 130 g/Day | 250 g/Day | 130 g/Day | 250 g/Day | 130 g/Day | 250 g/Day | 130 g/Day | 250 g/Day |
| Al | 1 mg/kg body weight/week [53] | TWI | 1.06 | 2.04 | 82.4 | 160 | 1.07 | 2.06 | 83.2 | 160 | 0.77 | 1.47 | 59.9 | 114 | 0.71 | 1.37 | 55.2 | 107 |
| B | 0.16 mg/kg body weight/day [54] | UL | 0.70 | 1.34 | 48.48 | 93.23 | 0.31 | 0.60 | 21.67 | 41.67 | 0.22 | 0.43 | 15.62 | 30.03 | 0.25 | 0.48 | 17.24 | 33.16 |
| Ba | 0.2 mg/kg bw/day [41] | TDI | 0.14 | 0.27 | 7.78 | 15.0 | 0.23 | 0.44 | 12.8 | 24.4 | 0.15 | 0.29 | 8.33 | 16.1 | 0.21 | 0.40 | 11.7 | 22.2 |
| Cd | 2.5 µg/kg bw/week [55] | TWI | 0.001 | 0.003 | 31.1 | 93.3 | 0.004 | 0.008 | 124 | 250 | 0.004 | 0.008 | 124 | 250 | 0.003 | 0.005 | 93.3 | 156 |
| Ni | 2.8 µg/kg bw/day [83] | TDI | 0.02 | 0.03 | 79.4 | 119 | 0.11 | 0.22 | 436 | 873 | 0.02 | 0.04 | 79.4 | 159 | 0.02 | 0.04 | 79.4 | 159 |
| Sr | 0.13 mg/kg bw/day [42] | | 0.46 | 0.89 | 39.3 | 76.1 | 0.56 | 1.09 | 47.9 | 93.2 | 0.34 | 0.66 | 29.1 | 56.4 | 0.53 | 1.03 | 45.3 | 88.0 |
| V | 1.8 mg/day [14] | UL | 0.02 | 0.04 | 1.16 | 2.22 | 0.02 | 0.05 | 1.37 | 2.64 | 0.02 | 0.04 | 1.01 | 1.94 | 0.02 | 0.03 | 0.94 | 1.81 |
| | Guideline Values | Parameter | EDI | | MOE | | EDI | | MOE | | EDI | | MOE | | EDI | | MOE | |
| | | | 130 g/day | 250 g/day | 130 g/day | 250 g/day | 130 g/day | 250 g/day | 130 g/day | 250 g/day | 130 g/day | 250 g/day | 130 g/day | 250 g/day | 130 g/day | 250 g/day | 130 g/day | 250 g/day |
| Pb | 0.5 µg/kg bw/day [56] | BMDL | 0.01 | 0.03 | 450 | 150 | 0.02 | 0.04 | 225 | 112.5 | 0.02 | 0.04 | 225 | 112.5 | 0.02 | 0.04 | 225 | 112.5 |

## 4. Consumption Recommendations

Considering the estimated daily intake values of both essential elements and toxic elements, a lower consumption of these products is recommended, avoiding daily consumption, since the intake of large portions and their frequent use can pose a risk to the health of the infants.

However, it is necessary to consider that children's compotes are an important source of essential elements such as Ca, Zn, Cu, Mg or Mo. For this reason, a series of recommendations to minimize exposure to the toxic metals analyzed are given in Figure 2.

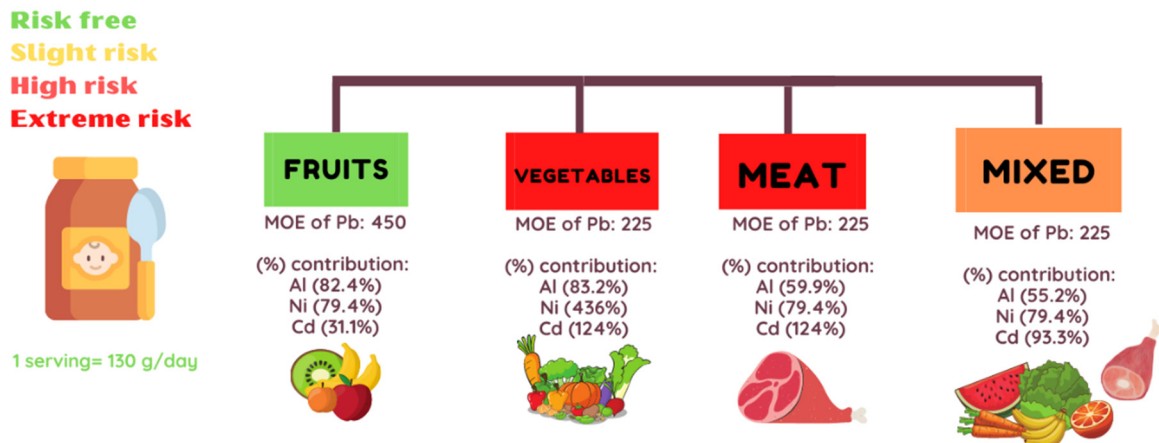

**Figure 2.** Baby food jar consumer guidelines.

It is recommended, in all cases, not to exceed 130 g/day of compote. In terms of the flavors analyzed, the consumption of fruit compotes is considered to be less risky as it is exposed to lower dietary contributions of toxic metals such as Pb (MOE of 450), Al (82.4% of the TWI), Ni (79.4% of the TDI) and Cd (31.1% of the TWI). Meanwhile, the consumption of mixed compotes is recommended but only sporadically, avoiding daily consumption as it offers percentages that are close to the reference values for toxic elements such as Cd (93.5% of the TWI). Finally, it is recommended to avoid the consumption of vegetable compotes or meat compotes, as the tolerable weekly and daily intake values for Cd (124%) or Ni (436%) are exceeded.

## 5. Conclusions

Mixed jars register the highest concentrations of essential elements such as Na, Ca, Zn and Fe. On the other hand, meat and vegetable jars recorded remarkable levels of K and Mg, respectively. The average levels of Li were high in the four types of baby food analyzed from jars. On the other hand, vegetable jars contain the highest levels of Al. The consumption of 130 g/day of mixed jars contribute to cover the daily requirements of Zn and Ca. Vegetable jars (130 g/day) represent the important contribution of Cu and Mg. However, both the proposed consumption scenarios (130 g/day and 250 g/day) assume very high percentages over the recommended values of most of the elements, highlighting Mn (intakes 40 times higher than the recommended value) or Mo (4 times higher than the maximum), and may pose a risk to the health of infants. The intake of toxic elements from the consumption of the analyzed baby food in jars (130 g/day and 250 g/day), such as Pb, Ni, Cd and Al, far exceeds the maximum values set by the different institutions consulted. The toxic effects of these elements are very disturbing, especially in sensitive individuals such as babies. Therefore, the two consumption scenarios of the infant jars analyzed are not safe for babies between the ages of 6 and 7–11 months. A low consumption of baby

food from jars is recommended, less than the amounts set out in this study, and spaced over time, avoiding daily consumption, as it could pose a risk to infants' health.

**Author Contributions:** Conceptualization, A.H. and Á.J.G.-F.; Formal analysis, S.G.-S., S.A.-V., D.G.-W. and Á.J.G.-F.; Funding acquisition, C.R.-A.; Investigation, S.G.-S., S.P.-M., D.G.-W. and A.H.; Methodology, S.G.-S., S.P.-M., D.N.-C., S.A.-V. and D.G.-W.; Resources, S.G.-S., D.N.-C., A.H. and Á.J.G.-F.; Supervision, A.H. and Á.J.G.-F.; Visualization, C.R.-A.; Writing—original draft, S.G.-S. and S.P.-M.; Writing—review and editing, S.P.-M., D.N.-C., S.A.-V. and C.R.-A. All authors have read and agreed to the published version of the manuscript.

**Funding:** This research received no external funding.

**Institutional Review Board Statement:** Not applicable.

**Informed Consent Statement:** Not applicable.

**Data Availability Statement:** Not applicable.

**Conflicts of Interest:** The authors declare that they have no conflict of interest.

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
