# Peer review of "Baby Food Jars as a Dietary Source of Essential (K, Na, Ca, Mg, Fe, Zn, Cu, Co, Mo, Mn) and Toxic Elements (Al, Cd, Pb, B, Ba, V, Sr, Li, Ni)"

_applsci, doi:10.3390/app12168044_

Round 1

Reviewer 1 Report

Dear Authors

Please find my comments below.

1) Regarding the wording of "baby food Jars" throughout the article is kind of misleading in the sense that it's the jars that contribute to the dietary requirement, not the food inside of them. My suggestion would be "baby food in jars"

2)  What is the chemical composition of jars and caps used in making those jars? Does the jar or any part of the lid is capable of leaching heavy elements into the food?

3) Does precision and accuracy are the only QC criteria you have used to validate the method?

4) While comparing to other studies, your values were consistently higher in all of the elements you have shown in table 6. Does your analytical method sensitivity play any role in this deviation?

5) Though you claim the reason for higher elemental concentration for your samples may be due to environmental pollution factors when comparing other author studies, those studies were done pretty recently (2017-2019). Please clarify.

Author Response

The response is on the attached document.

Thank you for your comments. 

Reviewer 2 Report

Santiago González-Suárez et al present the paper dedicated to the determination of essential and toxic elements in baby food jars. The research is important in a way of food consumption safety. The paper has some minor disadvantages.

Authors didn’t point out what new they contribute. ICP-OES is a well-known method for such a task. Any novelties concerning analytical method I didn’t find.

Please add the information and of course some references in the introduction section regarding the application of ICP-OES to the food products by other authors. The introduction part will be more interesting if you indicate the advantages of ICP-OES method comparing to other analytical methods.

The sections are wrongly numbered, please correct.

Have you observed any contaminations during the sample preparation procedure?

The analytical part is well-described, however authors forgot to indicate the calibration procedure. What materials you used as calibration samples? What is the concentration range etc.?

Author Response

Thank you for your comments. The response is on the attached document. 

Reviewer 3 Report

·         There is no reference to similar studies in the introduction. Obviously this is not the first study on this subject matter.

·         Authors needs to know that toxicity depends on body weight. Hence, the need to base their calculation on child’s age and body weight.

·         It is important that authors mention the city and country where the samples were collected.

·         Authors did not present individual results, instead average results of each category of the baby jar food were presented. This shouldn’t be because the composition of the samples is not the same, since they are not produced by the same company. It is possible that some of the baby jars will contain less toxic elements and a moderate or even adequate amounts of essential minerals.

·         It is expected that this study will advise baby food jar producers on how to reduce the levels of toxic and essential minerals in their products instead as opposed to advising on reduction in their intake. This is because these baby jar is easily accessible and affordable and hence busy mothers will prefer them.

·         Authors need to correct the grammatical errors. Maybe the article should be given to  an English-speaking person to proofread, or better still, authors can use Grammarly software.

Author Response

(The authors gave the same response as above.)

Round 2

Reviewer 1 Report

Thanks for addressing all my queries.